# Policy Adaptation via Language Optimization: Decomposing Tasks for Few-Shot Imitation

**Vivek Myers**[*], **Bill Chunyuan Zheng**[*], **Oier Mees, Sergey Levine**[†], **Kuan Fang**[†]
University of California, Berkeley

**Abstract:** Learned language-conditioned robot policies often struggle to effectively adapt to new real-world tasks even when pre-trained across a diverse set of instructions. We propose a novel approach for few-shot adaptation to unseen tasks that exploits the semantic understanding of task decomposition provided by vision-language models (VLMs). Our method, Policy Adaptation via Language Optimization (PALO), combines a handful of demonstrations of a task with proposed language decompositions sampled from a VLM to quickly enable rapid nonparametric adaptation, avoiding the need for a larger fine-tuning dataset. We evaluate PALO on extensive real-world experiments consisting of challenging unseen, long-horizon robot manipulation tasks. We find that PALO is able of consistently complete long-horizon, multi-tier tasks in the real world, outperforming state of the art pre-trained generalist policies, and methods that have access to the same demonstrations.[1]

**Keywords:** Few-shot Learning, Vision-Language Models, Robot Manipulation

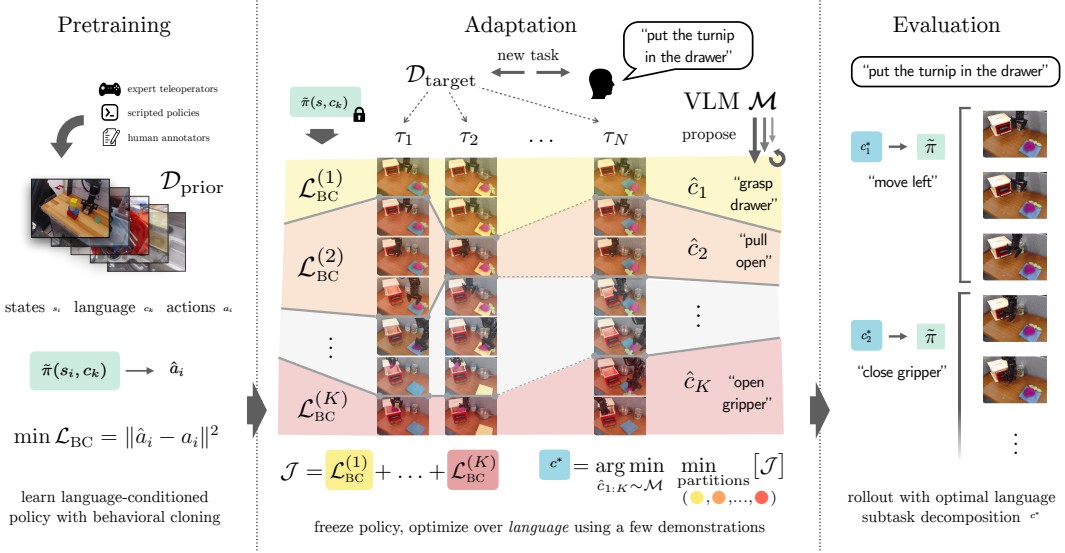

Figure 1: An overview of the PALO algorithm for few-shot adaptation with language. (Left) We build off a pretrained policy that has learned to follow low-level language instructions from a large dataset of expert demonstrations. (Middle) Given a new task and a few expert demonstrations, we use a VLM to propose candidate decompositions into subtasks. We optimize over these decompositions jointly with the partitions of trajectories into subtasks, selecting the subtask decomposition that minimizes the validation error of the learned policy. (Right) At test time, we condition the pre-trained policy on the selected decomposition to solve the task.

---

[*]Equal contribution.

[†]Equal advising.

[1]Project webpage: https://palo-website.github.io

8th Conference on Robot Learning (CoRL 2024), Munich, Germany.

# 1 Introduction

Robot learning promises policies that can adapt and generalize to new behaviors. However, in practice, today's robotic policies often struggle to effectively finetune for truly new tasks [1, 2, 3, 4, 5]. For example, consider the task of making a salad: while a person could likely follow a new recipe with only a few examples by remembering the key steps, a robot learning approach may need many more demonstrations to achieve similar performance, and still recover a more brittle policy.

A key difference that allows humans to learn tasks so quickly is their semantic understanding of the world. Human have a symbolic representation of the task, such as the names of the ingredients and the steps to prepare them, rather than a series of low-level actions. This representation enables them to understand the task at a higher level, mapping directly into low-level behaviors they are already familiar with [6, 7]. How can we enable robots to quickly learn new tasks through a semantic understanding of the world?

Language provides a potential bridge between these task semantics and low-level control [8]. Recent advances in large language models (LLMs) and vision-language models (VLMs) have shown promise in understanding and grounding language from a few demonstrations [9, 10]. We propose Policy Adaptation via Language Optimization (PALO), a method for exploiting the semantic understanding of VLMs in combination with a pre-trained robot policy to enable adaptation to new tasks with only a few demonstrations (Fig. 1).

Past approaches that fine-tune directly to new demonstrations are often overparameterized and sample-inefficient, due to the cost inherent in collecting teleoperated trajectories [11]. Instead, we use a few demonstrations as a calibration set to guide the decomposition of a new language task into a sequence of subtasks that can be used by a language-conditioned policy. Our approach samples possible decompositions of the task from a VLM and chooses the one that minimizes the validation error of the learned policy on the calibration set.

The key is that in the few-shot setting, a few demonstrations provide a better signal for

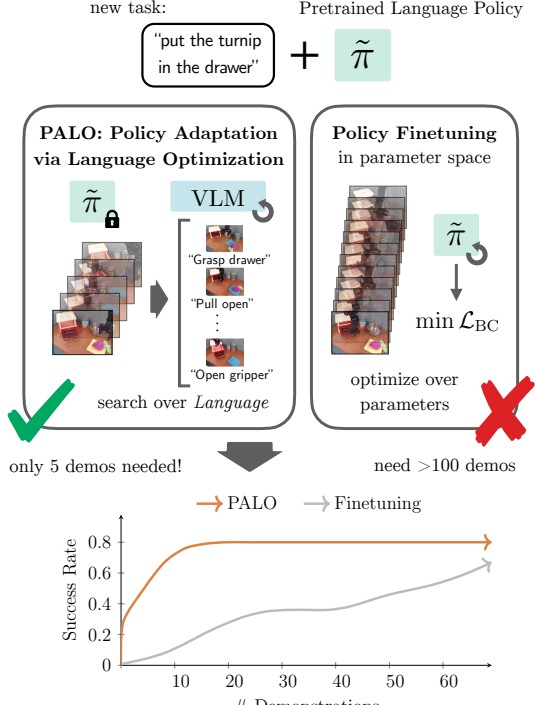

Figure 2: PALO enables pre-trained generalist policies to adapt new tasks with as few as five demonstrations by searching in language space instead of parameter space.

adapting to new tasks when used to select the right sequence of language subtasks with the help of a VLM, rather than directly fine-tuning the policy parameters (Fig. 2). Unlike prior work, our approach can learn unseen, long-horizon behaviors with fewer than 10 demonstrations across a variety of tabletop manipulation tasks.

# 2 Related Work

Our approach lies at the intersection of few-shot learning and approaches that leverage language and large pre-trained models for robotics.

**Few-shot learning.** Broadly speaking, few-shot learning approaches utilize diverse data to enable rapid test-time adaptation to a new task from a few examples. These techniques have been applied in various domains, including vision [12, 13, 14], natural language processing [9, 15], and reinforcement learning [16, 17]. Frameworks for few-shot learning include optimization-based meta-learning [18, 19, 20], where a model is trained to quickly fine-tune to new tasks, nonparametric

methods based on particular modeling assumptions such as metric approaches and Gaussian processes [21, 22, 23, 24], and in-context learning [9, 25, 26, 27], where a large model is conditioned on a context to adapt to a new task. Unlike past approaches to few-shot learning in robotics [28, 11], we show that language can be used to enable nonparametric adaptation without fine-tuning.

**Language-conditioned robotic control.** While early approaches to instruction-following in robotics relied on manually designed symbolic representations [29, 30, 31], recent work has focused on applying deep learning techniques to understand natural language instructions [32, 33]. These approaches use learned behavioral cloning policies on top of language [2, 34, 35], connect language representations to grounded representations of the environment [36, 37, 38, 39], or use the compositional structure of language to decompose tasks and plan [40, 41, 42, 43, 44]. Our approach is the first to enable few-shot adaptation to new demonstrations in robotics by leveraging the structure of language.

**Foundation models and robotics.** Large-scale internet pre-training has seen recent success in the domains of vision and natural language processing [9, 45, 46, 47, 9, 48]. Recent work has investigated if these models can be trained and/or fine-tuned for downstream robotics tasks [32, 49, 50, 11, 51, 52]. Other work has investigated if these models can be used to provide semantic knowledge for downstream robot learning pipelines [53, 54, 55, 56, 57, 58, 59]. Our approach falls into this latter category, but unlike the past works, we perform *few-shot adaptation* in language-conditioned robot control using the semantic knowledge in large pre-trained VLMs.

## 3  Policy Adaptation via Language Optimization

Our goal is to enable a learned language-conditioned robot policy to perform new tasks with only a few demonstrations. The key insight is that the structure of language can be exploited to enable few-shot adaptation to new demonstrations in robotics. Fundamentally, few-shot adaptation to new tasks depends on a policy's ability to generalize its existing knowledge to correctly fit to new demonstrations. One approach for adapting a learned policy is to directly fine-tune to new demonstrations, but in robotics settings where expert data collection is costly, this is often infeasible due to overfitting.

We propose Policy Adaptation via Language Optimization (PALO), which instead uses demonstrations of a task that is outside the training distribution with the reasoning capabilities of a pre-trained vision-language model (VLM) to determine the correct sequence of decomposed subtasks that are in-distribution for the robot policy. Given a language instruction $\ell$, we compute a task decomposition $c_{1:K}$ that is both semantically consistent with the instruction (determined by the VLM) and feasible in the environment (measured by policy validation loss on expert demonstrations).

### 3.1  Notation

Formally, we assume a contextual Markov Decision Process (MDP) structure. We have a state space $\mathcal{S}$, continuous action space $\mathcal{A} = (0,1)^{d_A}$, initial state distribution $p_0$, transition probabilities $P$, and free-form language instruction $\ell \in \mathcal{L}$ chosen from the language instruction space $\mathcal{L}$. We use the notation $\mathcal{P}(X)$ to denote the set of probability distributions over a space $X$.

The robot selects the action $a_t \in \mathcal{A}$ based on the observed state $s_t \in \mathcal{S}$ at each time step $t \in \{1 \ldots H\}$ over a finite horizon $H$ to achieve states in $\mathcal{S}_\ell$. We denote a robot policy as a map $\pi(\hat{a}_t \mid s_t, \ell)$, which maps the state $s_t$ and instruction $\ell$ to a distribution over actions $\hat{a}_t$. For convenience, we assume actions are selected under a fixed isotropic Gaussian noise model unless otherwise specified, and will denote the mode of the distribution $\pi(\hat{a} \mid s_t, \bullet)$ as $\pi(s_t, \bullet)$. A robot policy then yields a distribution over trajectories $\left( \{(s_i, a_i)\}_{i=1}^H, \ell \right) \sim \mathcal{T}_\pi^\rho$ given a task distribution $\rho \in \mathcal{P}(\mathcal{L})$.

### 3.2  Problem Statement

We want to solve out-of-distribution instruction-following tasks involving unseen objects and skills given only a few demonstrations. For (pre-)training the instruction-following policy we assume access to a dataset that has been generated using language tasks sampled from some distribution $\rho_{\text{prior}} \in \mathcal{P}(\mathcal{L})$ with an expert policy $\pi_\beta(\hat{a} \mid s, \ell)$. For training an instruction-following policy $\hat{\pi}(s, \ell)$, we assume a prior dataset $\mathcal{D}_{\text{prior}} = \left\{ (\tau^{(i)}, c_{1:K}^{(i)}, \ell^{(i)}) \right\}_{i=1}^{N_{\text{prior}}}$ for $\tau^{(i)}, \ell \sim \mathcal{T}_{\pi_\beta}^{\rho_{\text{prior}}}$ and additional

hierarchically-decomposed subtask instructions $c_{1:K} \in \mathcal{L}^K$ that are distributed according to $p(c_{1:K} \mid s_0, \ell)$ for decomposition size $K < H$.

A target task is sampled from a separate distribution $\rho_{\text{target}} \in \mathcal{P}(\mathcal{L})$ which requires interacting with unseen objects in novel ways, so the policy trained on $\mathcal{D}_{\text{prior}}$ performs poorly zero-shot. To solve this new task, we assume there exists an additional dataset $\mathcal{D}_{\text{target}} = (\{\tau_1 \dots \tau_n\}, \ell)$ for $\tau_i \sim \mathcal{T}_{\pi_\beta}^{\delta_\ell} \mid s_0$ with $s_0 \sim p_0$ and $\ell \sim \rho_{\text{target}}$ collected by human experts. While a large $\mathcal{D}_{\text{target}}$ can enable directly training $\pi(s, \ell)$ to solve the target task, we are interested in challenging few-shot scenarios in which $\mathcal{D}_{\text{target}}$ only contains a handful of demonstrations (e.g., 5). In this paper, we tackle this challenge by decomposing the novel target task into a sequence of subtasks that are solvable by the pre-trained $\tilde{\pi}(s, c)$ using a VLM $\mathcal{M}$. Notably, we do not assume any ground truth labels for the task decomposition are given, and aim to generate the optimal language decomposition $c_{1:K}$ based on the unlabeled demonstration dataset $\mathcal{D}_{\text{target}}$ collected by human operators.

Our approach makes two assumptions about the structure of the target task.

**Assumption 1.** *The target task subtask annotations $c_i$ locally match those of the prior dataset, i.e., are distributed identically for $i \sim \text{Unif}(1 \dots H)$*

$$\mathbb{E}_{\ell \sim \rho_{\text{target}}, s_0 \sim p_0} p(c_i \mid s_0, \ell) \approx \mathbb{E}_{\ell \sim \rho_{\text{prior}}, s_0 \sim p_0} p(c_i \mid s_0, \ell). \tag{1}$$

Assumption 1 states that even if the overall target tasks in $\rho_{\text{target}}$ are unseen, the low-level manipulation skills (e.g., "close the gripper," "move the arm right") will be represented in the policy training.

**Assumption 2.** *The VLM $\mathcal{M}$ can approximate the distribution of the subtask annotations $c_{1:K}$ in the target task, i.e.,*

$$p_{\mathcal{M}}(c_{1:K} \mid s_0, \ell) \approx p(c_{1:K} \mid s_0, \ell). \tag{2}$$

Assumption 2 states that the VLM can propose candidate task decompositions that are consistent with the instruction $\ell$ in new scenes. Qualitatively, these assumptions are consistent with recent advances in robot manipulation training data [5, 51] and embodied reasoning with VLMs [60] and are empirically validated in our experiments in Section 4 using the BridgeDataV2 dataset [5] and GPT-4o [46] with prompting described in Appendix F.

In Section 3.6 we show that under these assumptions, our PALO algorithm can achieve low regret on out-of-distribution tasks, and discuss how violating these assumptions affects performance.

### 3.3 Task Decomposition with Language

To guide the pre-trained policy $\hat{\pi}$ to solve the unseen target task, we decompose the high-level language instruction $\ell$ of the target task into a sequence of subtask instructions $c_{1:K} = (c_1, \dots, c_K)$ for the $K$ subtasks as a set of language decomposition. Instead of commanding $\hat{\pi}$ with the original instruction $\ell$, we use a combination of $\ell$ and the subtask instructions $c_k$ as the input in each subtask to produce the action as $a_t \leftarrow \tilde{\pi}(s_t, c_k)$. In our methods, we used GPT-4o [46] as a backbone to generate instruction sets. We denote by $\mathcal{M}(s_0, \ell)$ the support of possible task decompositions sampled from this VLM (see details in Appendix F).

Aside from the sequential order of the subtasks, the robot needs to decide when to switch to the next subtask. For this purpose, we introduce an additional variable $u = (u_1, \dots, u_K) \sim \text{Unif}(\mathcal{U})$ where $\mathcal{U}$ is the space of ordered partitions of $\{0 \dots H\}$, so $u_k$ denotes the time steps on which the robot is executing the $k$-th subtask. Notably, we assume the optimal solution to the target task follows a fixed structure, i.e., the same subtask sequence $c$ can be used to solve the task, regardless of the initial state $s_0$. Meanwhile, $u$ can be different in each episode, since the number of steps needed to complete each subtask depends on $s_0$ as well as stochasticity in the environment and the policy.

### 3.4 Few-Shot Adaptation through Language Decomposition

We design a simple sampling-based inference algorithm to find the best $c^*$ for guiding the policy $\tilde{\pi}$ to solve the target task. Since the resulting actions depend on both $c$ and $u$, as discussed in Section 3.3, we jointly optimize $c$ and $u$ to minimize a cost function $\mathcal{J}$ over all trajectories in $\mathcal{D}_{\text{target}}$:

$$\min_{c_{1:K} \in \mathcal{M}(s_0, \ell)} \sum_{\tau \in \mathcal{D}_{\text{target}}} \left( \min_{u_{1:K} \in \mathcal{U}} \mathcal{J}(c, u, \tau) \right). \tag{3}$$

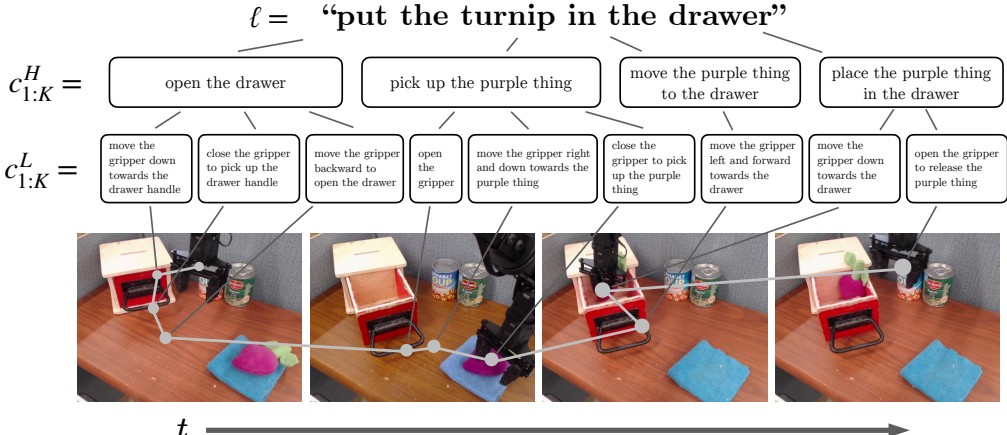

Figure 3: A visualization of an example execution of our method on the long-horizon task "put the beet toy in the drawer." The VLM deconstructs $\ell$ into candidate high-level subtasks $c_{1:K}^H$ and low-level subtasks $c_{1:K}^L$ and optimizes over the expert trajectories. The optimal $c_{1:K}^H$ and $c_{1:m}^L$ are chosen and unrolled in real-world evaluations, which result in successful completion of the task (trajectory shown in gray).

To measure how well $c$ and $u$ enable the policy $\tilde{\pi}$ to reproduce each $\tau$, the cost function is defined with the mean squared error between the predicted action $\hat{a}_t$ and the ground truth $a_t$ at each time step $t$. More specifically, we evaluate the policy $\tilde{\pi}$ on the demonstration trajectory given $c$ and $u$ to compute $\hat{a}_t \leftarrow \tilde{\pi}(s_t, c_{\min\{k:t \in u_k\}})$. Then the cost function is defined as:

$$\mathcal{J}(c, u, \tau) = \sum_{n=1}^{K} \sum_{t \in u_n} \left\| a_t - \tilde{\pi}(s_t, c) \right\|^2. \tag{4}$$

By minimizing this cost across demonstrations, we compute a decomposition of the task $c$ that would optimally perform the task by minimizing the loss between the action of the robot and the expert.

### 3.5 Learning Composable Instruction-Following Primitives

We use language-conditioned behavior cloning [61] to learn a policy $\hat{\pi}(s_t, \ell)$ based on the expert trajectories of $\mathcal{D}_{\text{prior}}$. To enable conditioning on fine-grained hierarchical language instructions, we factorize $\hat{\pi}$ through $c_{1:K}$:

$$\hat{\pi}(\hat{a} \mid s_t, \ell) = \sum_{c_{1:K} \in \mathcal{L}} p(c_{1:K} \mid \ell) \sum_{k=1}^{K} \tilde{\pi}(\hat{a} \mid s_t, c_k) p(\mathbf{k}_t = k) \tag{5}$$

for the subtask index at time $t$: $\mathbf{k}_t = \min\{k : t \in u_k, u_{1:K} \sim \text{Unif}(\mathcal{U})\}$. We learn parameters $\theta$ for $\tilde{\pi}_\theta$ by minimizing the following behavioral cloning objective:

$$\mathcal{L}_{\text{BC}}(\theta) = \mathbb{E}_{(s_t, a_t, c_k, \ell) \sim \mathcal{D}_{\text{prior}}} \left[ \sum_{t=1}^{H} \left\| \tilde{\pi}_\theta(s_t, c_k) - a_t \right\|^2 \right]. \tag{6}$$

The training dataset $\mathcal{D}_{\text{prior}}$ is an augmented version of BridgeData [5], a dataset containing a diverse set of manipulation tasks on common household objects. Details about how the subtask instructions are generated are discussed in Appendix D. Each $c_i$ is further partitioned into a high-level component $c_i^H$ and a low-level component $c_i^L$. Our full implementation is described in Appendix D.

### 3.6 Analysis of PALO

Our theoretical results study the regret of this approach on out-of-distribution tasks in $\rho_{\text{target}}$, showing that it trades off the performance of the pre-trained policy on $\rho_{\text{prior}}$ and the performance of the VLM $\mathcal{M}$ in accurately modeling the hierarchical language decomposition $p(c_{1:K})$ in $\rho_{\text{target}}$. We define regret with respect to the expert policy $\pi_\beta$ and a given task distribution in terms of the MSE:

$$R_{\pi_\beta}(\pi; \rho) = \mathbb{E}_{\mathcal{T}_{\pi_\beta}^\rho} \left[ \frac{1}{H\sqrt{d_A}} \sum_{t=1}^{H} \left\| \pi(s_t, \ell) - \pi_\beta(s_t, \ell) \right\|^2 \right]. \tag{7}$$

**Algorithm 1** Policy Adaptation via Language Optimization (PALO)

---

**Require:** a VLM $\mathcal{M}$, pre-trained instruction-following policy $\pi(\hat{a} \mid s_t, c)$,
number of candidate decompositions to sample $M$, optimization steps $N$
**Input:** new task described by $\ell$ with $n$ expert demonstrations $\mathcal{D}_{\text{target}}$ collected manually
**Output:** policy $\hat{\pi}(\cdot \mid s_t)$ adapted to the new task $\ell$

1: **for** $i = 1$ to $M$ **do**
2: $\quad c_{1:K}^{(i)} \sim \mathcal{M}(s_0, \ell)$
3: $\quad$ **for** $j = 1$ to $N$ **do**
4: $\quad\quad u_{1:K}^{(i,j)} \sim \text{Unif}(\mathcal{U})$
5: $\hat{c}_{1:K} \leftarrow \arg\min_{c_{1:K} \in \{c^{(i)}\}_{i=1}^M} \min_{u \in \{u^{(i,j)}\}_{j=1}^N} \mathcal{J}(c_{1:K}, u, \tau)$ $\qquad\qquad$ (eq. 4)
6: $\pi_{\text{PALO}}(\hat{a} \mid s_t, \ell) \leftarrow \pi(\hat{a} \mid s_t, \hat{c}_{\mathbf{k}_t})$
7: **return** $\pi_{\text{PALO}}$.

---

**Theorem 3.7.** *The (out-of-distribution) regret of* PALO *on* $\rho_{\text{target}}$ *can be bounded as:*

$$R_{\pi_\beta}(\pi_{\text{PALO}}; \rho_{\text{target}}) \leq R_{\pi_\beta}(\hat{\pi}; \rho_{\text{prior}}) + \mathbb{E}\big[D_{\text{TV}}\big(p_{\text{target}}(c_{\mathbf{k}_t}), p_{\text{prior}}(c_{\mathbf{k}_t})\big)\big]$$

$$+ \big(2D_{\text{KL}}\big[p(c_{1:K}), p_{\mathcal{M}}\big]\big)^{1/2} + \frac{\sqrt{M} + \sqrt{n\log(Mn)}}{n} + 1/M + 1/K + N^{-2/K} \quad (8)$$

*where* $\pi_{\text{PALO}}$ *is from Algorithm 1,* $\hat{\pi}(s_t, \ell)$ *is trained on* $\mathcal{D}_{\text{prior}}$ *(Section 3.5), and* $t \sim \text{Unif}(1 \ldots H)$.

The proof is in Appendix A. Theorem 3.7 shows that in the limit as $N, M \to \infty$, we can decompose the out-of-distribution regret of PALO into a sum of the in-distribution regret of the pre-trained policy, and the performance of the VLM in accurately decomposing language tasks:

$$R_{\pi_\beta}(\pi_{\text{PALO}}; \rho_{\text{target}}) \lesssim \underbrace{R_{\pi_\beta}(\hat{\pi}; \rho_{\text{prior}})}_{\text{pre-training MSE}} + \underbrace{\big(2\,\mathbb{E}_{\rho_{\text{target}}} D_{\text{KL}}\big[p(c_{1:K})\big\|p_{\mathcal{M}}(c_{1:K})\big]\big)^{1/2}}_{\text{VLM accuracy}} + \underbrace{\mathbb{E}\big[D_{\text{TV}}\big(p_{\text{target}}(c_{\mathbf{k}_t}), p_{\text{prior}}(c_{\mathbf{k}_t})\big)\big]}_{\text{local marginal conformity}}. \quad (9)$$

Viewing the **VLM accuracy** and **local marginal conformity** terms as the extent to which Assumptions 1 and 2 are satisfied, we can see that under these conditions, Theorem 3.7 lets us directly relate the performance of the pre-trained policy $\hat{\pi}$ on the training data $\mathcal{D}_{\text{prior}}$ to the performance of the PALO algorithm on out-of-distribution tasks.

### 3.8 System Details

We use a ResNet-34 [62] to model the policy $\pi(a \mid s, c)$, where $c = (c^H, c^L)$ is a concatenation of high- and low-level instructions. The instruction $c = (c^L, c^H)$ is passed through a frozen MUSE model [63] to obtain embeddings before being fused into the ResNet with FiLM layers [64]. Architecture details are presented in Appendix C, and the overall algorithm is shown in Algorithm 1.

## 4 Experiments

In this section, we show that PALO can better adapt to long-horizon and out-of-distribution tasks from a few expert demonstrations than existing learned language-conditioned manipulation policies (both zero-shot and when finetuned to demonstrations), as well as a nonparametric few-shot adaptation method. Ablation studies also show all components of PALO are necessary.

### 4.1 Experimental Setup

We evaluate on a variety of long-horizon and/or unseen tasks across four scenes from the Bridge tabletop manipulation setup [5]. These involve manipulating new combinations of objects and behaviors unseen in the training data to accomplish long-horizon tasks, such as making a salad or pouring into a bowl. For each task, we collect a set of five expert demonstrations $\mathcal{D}_{\text{target}}$ for few-shot learning. Besides separating by scenes, we can also separate the tasks into 4 long-horizon tasks (put in, salad, sweep mints, sweep skittles) and 4 unseen-skills tasks (pry away, pour spoon, rotate marker, rotate spoon). Experimental details and example rollouts are presented in Appendix B.

### 4.2 Baselines

We compare against the following baselines trained on BridgeData:

**Octo [11]:** A general transformer-based robot manipulation policy with diffusion action head.

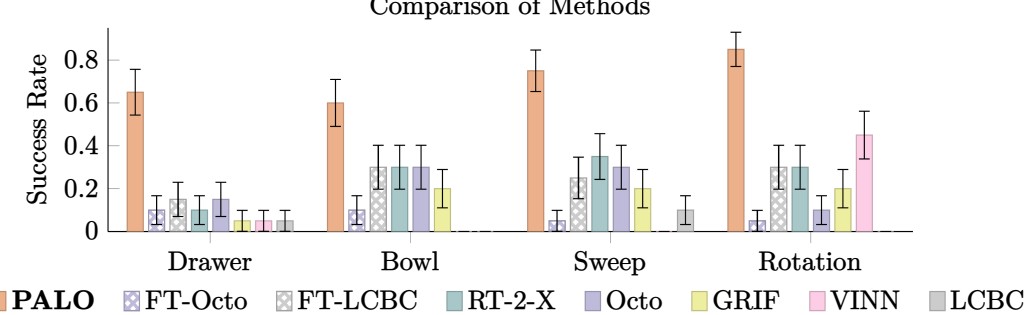

Figure 4: Comparison of PALO with baseline methods on different scenes with one standard error.

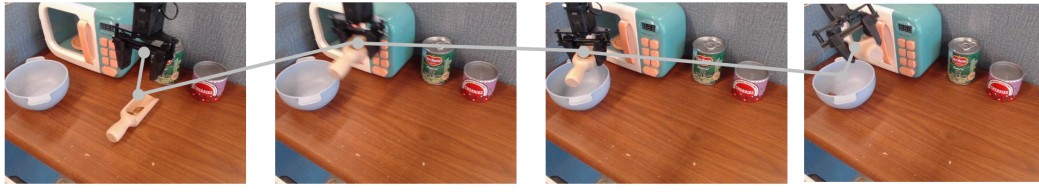

Figure 5: An execution of our method on the task "pour the contents of the scoop into the bowl." Full breakdown of task and instructions can be seen at Appendix G

**GRIF [36]:** A language-conditioned robot control method that uses pre-trained CLIP [45] representations to connect language instructions to goals for the policy to reach.

**RT-2-X [32]:** A language-conditioned robot control model with 55B parameters that transfers knowledge from internet-scale pre-training to manipulation zero-shot.

**LCBC [61]:** Language imitation with a ResNet and pretrained MUSE [63] embeddings.

**VINN [65]:** Using k-Nearest Neighbor to select actions from the training data based on similarity between the task representations of the observation and training data. We used GRIF's CLIP encoder for the representations used to calculate similarity scores.

**FT-Octo:** Octo transformer finetuned on the few-shot demonstration (see Appendix C.2 for details).

**FT-LCBC:** Similar to FT-Octo, but fine-tuning LCBC on the few-shot demonstrations.

Across eight different tasks, our PALO method yielded a success rate of 71.3%, while the best zero-shot policies only resulted in a success rate of 26.3%. While most of the zero-shot methods degrade when the task became increasingly more out-of-distribution for the pretrained policy (for example, tasks in the "salad" scene achieved a 30% overall performance across the 4 baseline models while pouring from scoop only achieved 12% performance across the models), our method remained effective, with all 8 tasks performing at a success rate of 50% or better.

The **FT-Octo** and **FT-LCBC** baselines allow us to compare the nonparametric adaptation of PALO to conventional parametric finetuning. While Octo trained only on BridgeData achieved moderate zero-shot success, finetuning on only five demonstrations overfit and worsened performance. The FT-LCBC baseline did benefit from finetuning, but still failed to ever exceed 30% success rate across all tasks. We observe that the small size of trajectories made these datasets an unfavorable candidate for finetuning, as any variance brought by the human controller may be amplified and cause unfavorable movements during evaluation. The nonparametric **VINN** baseline performed well on the rotation tasks (45% success rate), but failed to achieve greater than 5% success rate on the other tasks.

### 4.3 Ablations

We ablate the following components of our method in Fig. 6:

**Ours:** Our full PALO approach
**No $c^H$:** No high-level $c_H$ conditioning for the learned policy via masking.
**No $c^L$:** No low-level $c_L$ instruction conditioning via masking.

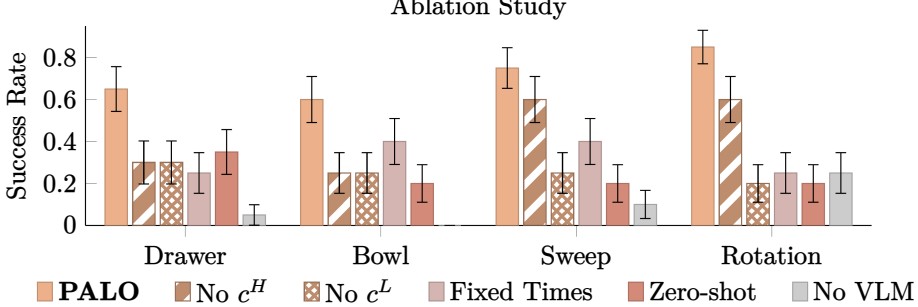

Figure 6: Ablation study of PALO on different scenes, plotted with one standard error.

**Fixed Times:** Use fixed $u = [\frac{H}{k}, \frac{2H}{k}, \dots, \frac{(k-1)H}{k}]$ in each trajectory to evaluate Eq. (3).
**Zero-Shot Decomposition:** Generate $c$ zero-shot without expert demonstrations.
**No VLM:** No VLM decomposition proposals by using only $\ell$ with our policy.

While the sweeping and rotation scenes gave comparable performance with masked high level instructions (**No** $c^H$), the performance deteriorated in Drawer and Bowl, which involved more unfamiliar items for the pretrained policy. The remaining ablations (**No** $c^L$, **Fixed Times**, **Zero-Shot Decomposition**, **No VLM**) decreased performance across all scenes. These approaches are discussed in more depth in Appendix E. Overall results in Fig. 6 show that all components of the PALO method are needed.

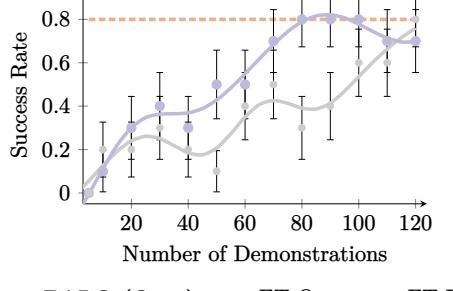

### 4.4 Scaling with Demonstrations

We study the scaling of our nonparametric method and a parametric finetuning approach with $> 5$ demonstrations of the skittle sweep-

Figure 7: Performance of PALO with 5 demonstrations compared to finetuning Octo on different number of demonstrations, plotted with one standard error.

ing task in Fig. 7. We observe that while Policy Adaptation via Language Optimization achieves the best performance (80%) using any number of demonstrations, the Octo finetuning baseline needs at least 80 expert demonstrations to achieve comparable performance, while LCBC needs at least 120 demonstrations.

### 4.5 Qualitative Results

We show successful task executions in Figs. 3 and 5. While the full method is robust to logically unsound instructions generated by the VLM, failures in reasoning and execution occur when we ablate our methods. Fig. 12 and Fig. 13 are two examples in which reasoning break down in ablations. See Appendix G for details.

## 5 Discussion

We introduced PALO, an approach for few-shot adaptation to unseen tasks that exploits the semantic understanding of task decomposition provided by vision-language models. In extensive real world experiments, we find that PALO is able to use language to adapt to unseen long-horizon robot manipulation tasks across a wide range of tabletop setups.

**Limitations and Future Work.** We assume the dataset has a consistent format of high-level language labels and proprioception, making it more challenging to generalize our low-level heuristic generation on drastically different embodiments. The discrete optimization over subtask time steps may also scale poorly with the number of subtasks and time steps. Future work could explore more efficient optimization methods for this problem.

## Acknowledgements

This research was partly supported by AFOSR FA9550-22-1-0273, ARO W911NF-21-1-0097, NSF IIS-2246811, DARPA ANSR, and the DoD through the NDSEG Fellowship Program.

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

## A   Proof of Theorem 3.7

**Theorem 3.7.** *The (out-of-distribution) regret of* PALO *on* $\rho_{\text{target}}$ *can be bounded as:*

$$R_{\pi_\beta}\left(\pi_{\text{PALO}}; \rho_{\text{target}}\right) \leq R_{\pi_\beta}(\hat{\pi}; \rho_{\text{prior}}) + \mathbb{E}\left[D_{\text{TV}}\left(p_{\text{target}}(c_{\mathbf{k}_t}), p_{\text{prior}}(c_{\mathbf{k}_t})\right)\right]$$

$$+ \left(2D_{\text{KL}}\left[p(c_{1:K}), p_{\mathcal{M}}\right]\right)^{1/2} + \frac{\sqrt{M} + \sqrt{n \log(Mn)}}{n} + 1/M + 1/K + N^{-2/K} \quad (8)$$

*where* $\pi_{\text{PALO}}$ *is from Algorithm 1,* $\hat{\pi}(s_t, \ell)$ *is trained on* $\mathcal{D}_{\text{prior}}$ *(Section 3.5), and* $t \sim \text{Unif}(1 \ldots H)$.

*Proof.* We will first consider the empirical regret of the MLE estimate of $c_{1:K}$, and relate it to in-distribution regret of $\tilde{\pi}$ using PAC techniques (see Catoni [66], Alquier [67]). We will then bound the remaining error due to the approximations made by the PALO algorithm and this empirical regret.

Recall our definition of regret:

$$R_{\pi_\beta}(\pi; \rho) = \mathbb{E}_{\mathcal{T}_{\pi_\beta}^\rho}\left[\frac{1}{H\sqrt{d_A}} \sum_{t=1}^{H} \left\|\pi(s_t, \ell) - \pi_\beta(s_t, \ell)\right\|^2\right]. \qquad \text{(from eq. 7)}$$

We can also define an empirical target regret $R_{\text{EMP}}$ measuring the fit of some $c \in \mathcal{L}^K$ to the target distribution $\rho_{\text{target}}$ in terms of Eq. (4):

$$R_{\text{EMP}}(c_{1:K}) = \mathbb{E}_{\mathcal{D}_{\text{target}} \sim \rho_{\text{target}}}\left[\frac{1}{H\sqrt{d_A}} \sum_{\tau \in \mathcal{D}_{\text{target}}} \min_{u_{1:K}} \mathcal{J}(c_{1:K}, u_{1:K}, \tau)\right] \qquad (10)$$

where $\mathcal{J}$ is the cost function in Eq. (4). PALO selects $c_{\text{PALO}} = \arg\min_{c_{1:K} \in \mathcal{L}^K} \widehat{R}_{\text{EMP}}(c_{1:K})$ to minimize an approximation of this quantity for samples $u^{(1)}, \ldots, u^{(N)} \sim \text{Unif}(\mathcal{U})$:

$$\widehat{R}_{\text{EMP}}(c_{1:K}) = \mathbb{E}_{\mathcal{D}_{\text{target}} \sim \rho_{\text{target}}}\left[\frac{1}{H\sqrt{d_A}} \sum_{\tau \in \mathcal{D}_{\text{target}}} \min_{i \in \{1 \ldots N\}} \mathcal{J}(c_{1:K}, u^{(i)}, \tau)\right]. \qquad (11)$$

We will also define a distributional notion of conditional regret for our analysis:

$$\widetilde{R}_{\pi_\beta}\left(\tilde{\pi} \mid s_0, \ell, c_{1:K}\right) = \mathbb{E}_{\tau \sim \mathcal{T}_{\pi_\beta}^{s_0}}\left[\frac{1}{H\sqrt{d_A}} \min_{u_{1:K} \in \mathcal{U}} \mathcal{J}(c_{1:K}, u_{1:K}, \tau)\right]. \qquad (12)$$

We now make use of the following PAC result [68, 67], which follows from Hoeffding's inequality:

**Lemma A.1** (Alquier [67, Theorem 1.2]). *Let* $\mathcal{H}$ *be a class of functions* $f : \mathcal{X} \to [0,1]$ *with* $|\mathcal{H}| = M$, *and let* $\rho \in \mathcal{P}(X)$ *be an arbitrary data distribution. Further, suppose* $\mathcal{D}$ *is a sample of size* $n$ *drawn i.i.d. from* $\rho$. *Then, for any* $\varepsilon \in (0, 1)$, *we have*

$$\Pr\left(\forall f \in \mathcal{H}, \underbrace{\mathbb{E}_{x \sim \rho}[f(x)]}_{\text{generalization risk}} \leq \underbrace{\mathbb{E}_{x \sim \mathcal{D}}[f(x)]}_{\text{empirical risk}} + \sqrt{\frac{\log M - \log \varepsilon}{2n}}\right) \geq 1 - \varepsilon. \qquad (13)$$

Taking $X$ to be the space of trajectories and $\mathcal{H} = \mathcal{M}(s_0, \ell)$ for $f(c) = \min_u \mathcal{J}(c, u, \tau)$, we can apply Lemma A.1 to the empirical regret $R_{\text{EMP}}$ in Eq. (10) to obtain (for any $\varepsilon \in (0, 1)$)

$$\Pr\left(\forall c_{1:K} \in \mathcal{M}(s_0, \ell), R_{\pi_\beta}\left(\hat{\pi} \mid s_0, \ell, c_{1:K}\right) \leq R_{\text{EMP}}(c_{1:K}) + \sqrt{\frac{\log M - \log \varepsilon}{2n}}\right) \geq 1 - \varepsilon. \qquad (14)$$

Taking $c_{\text{PALO}}$ to be the output of the PALO algorithm, we can relate the true regret of PALO on the current task (left) to its empirical regret (right):

$$\Pr\left(R_{\pi_\beta}\left(\hat{\pi} \mid s_0, \ell, c_{\text{PALO}}\right) \le R_{\text{EMP}}\left(c_{\text{PALO}}\right) + \sqrt{\tfrac{\log M - \log \varepsilon}{2n}}\right) \ge 1 - \varepsilon. \tag{15}$$

Since regret is bounded by 1, we can convert to an expectation:

$$\mathbb{E}_{\mathcal{D}_{\text{target}}}\left[R_{\pi_\beta}\left(\hat{\pi} \mid s_0, \ell, c_{\text{PALO}}\right)\right] \le \mathbb{E}_{\mathcal{D}_{\text{target}}}\left[R_{\text{EMP}}\left(c_{\text{PALO}}\right)\right] + \sqrt{\tfrac{\log M - \log \varepsilon}{2n}} + \varepsilon.$$

**Lemma A.2.** *Suppose $u, u' \sim \text{Unif}(\mathcal{U})$ are i.i.d. samples from a uniform distribution over the ordered $K$-partitions $\mathcal{U}$ of $\{1 \dots H\}$. For any $\varepsilon \in [0, 1/K]$, we have*

$$\Pr\left(\textstyle\sum_{k=1}^{K} |u_k \cap u'_k| \le H\varepsilon\right) \le e^{-2H(\frac{1}{K} - \varepsilon)^2}.$$

**Lemma A.3.** *There exists an $\varepsilon \in [0, 1/K]$ such that*

$$\varepsilon + e^{-2H(\frac{1}{K} - \varepsilon)^2} \le 1/K + N^{-2/K}. \tag{16}$$

Since Algorithm 1 (line 4) only samples $N$ values for $u$ instead of the full space for the min in Eq. (10), we must separately consider the degree of suboptimality in the decomposition $c_{\text{PALO}}$ relative to the optimal $c^* = \arg\min_{c \in \mathcal{M}(s_0, \ell)} R_{\text{EMP}}(c)$ that results from our approach to determine the effect of $N$ on the final bound. Applying Lemma A.2, we can say:

$$\mathbb{E}_{\mathcal{D}_{\text{target}}}\left[\widetilde{R}_{\pi_\beta}\left(\hat{\pi} \mid s_0, \ell, c_{\text{PALO}}\right)\right]$$

$$\le \mathbb{E}_{\mathcal{D}_{\text{target}}}\left[R_{\text{EMP}}\left(c_{\text{PALO}}\right)\right] + \sqrt{\tfrac{\log M - \log \varepsilon}{2n}}$$

$$\le \mathbb{E}_{\mathcal{D}_{\text{target}}}\left[R_{\text{EMP}}\left(c^*\right)\right] + \sqrt{\tfrac{\log M - \log \varepsilon}{2n}} + \varepsilon + e^{-2H(\frac{1}{K} - \varepsilon)^2}$$

$$\le \mathbb{E}_{\mathcal{D}_{\text{target}}}\left[R_{\text{EMP}}\left(c^*\right)\right] + \sqrt{\tfrac{\log M - \log \varepsilon}{2n}} + 1/K + N^{-2/K}.$$

For $\varepsilon = \sqrt{M}/n$, we get

$$\mathbb{E}_{\mathcal{D}_{\text{target}}}\left[R_{\pi_\beta}\left(\hat{\pi} \mid s_0, \ell, c_{\text{PALO}}\right)\right]$$

$$\le \mathbb{E}_{\mathcal{D}_{\text{target}}}\left[R_{\text{EMP}}\left(c^*\right)\right] + \frac{\sqrt{M} + \sqrt{n\log(Mn)}}{n} + 1/K + N^{-2/K}. \tag{17}$$

So, we have related the true regret of PALO on the current task (left) to its empirical regret in the limit of infinite samples (right). All that remains is to compute the empirical regret, for which we make use of the following lemmas.

**Lemma A.4.** *Denote the true (unobserved) target decomposition as $c_{1:K}$. We can relate the empirical regret of the optimal PALO solution $c^*$ to the empirical regret of the true decomposition.*

$$\mathbb{E}_{\mathcal{D}_{\text{target}}}\left[R_{\text{EMP}}\left(c^*\right)\right] \le \mathbb{E}_{\mathcal{D}_{\text{target}}}\left[R_{\text{EMP}}\left(c_{1:K}\right) + D_{\text{TV}}\left(p(c_{1:K}), p_{\mathcal{M}}(c_{1:K})\right)\right] + 1/M$$

**Lemma A.5.** *The empirical regret of $\tilde{\pi}$ can be bounded for $i \sim \text{Unif}(1 \dots K)$ as*

$$\mathbb{E}_{\mathcal{D}_{\text{target}}}\left[R_{\text{EMP}}\left(c_{1:K}\right)\right] \le R_{\pi_\beta}(\hat{\pi}; \rho_{\text{prior}}) + \mathbb{E}\left[D_{\text{TV}}\left(p_{target}(c_{\mathbf{k}_t}), p_{prior}(c_{\mathbf{k}_t})\right)\right].$$

Applying Lemma A.4 and Lemma A.5 to Eq. (17) yields a bound of the correct form.

$$\mathbb{E}_{\mathcal{D}_{\text{target}}}\left[R_{\text{EMP}}\left(c^*\right)\right] \le \mathbb{E}_{\mathcal{D}_{\text{target}}}\left[R_{\text{EMP}}\left(c_{1:K}\right)\right] + D_{\text{TV}}\left(p(c_{1:K}), p_{\mathcal{M}}\right) + 1/M$$

$$\le \mathbb{E}_{\mathcal{D}_{\text{prior}}}\left[R_{\pi_\beta}(\hat{\pi}; \rho_{\text{prior}})\right] + \mathbb{E}\left[D_{\text{TV}}\left(p_{\text{target}}(c_{\mathbf{k}_t}), p_{\text{prior}}(c_{\mathbf{k}_t})\right)\right] + D_{\text{TV}}\left(p(c_{1:K}), p_{\mathcal{M}}\right) + 1/M.$$

To make the $D_{\text{TV}}\left(p(c_{1:K}), p_{\mathcal{M}}\right)$ term more interpretable as a VLM accuracy, we convert to a KL divergence with Pinsker's inequality [69]:

$$\mathbb{E}_{\mathcal{D}_{\text{target}}}\left[R_{\text{EMP}}\left(c_{\text{PALO}}\right)\right] \le \mathbb{E}_{\mathcal{D}_{\text{prior}}}\left[R_{\pi_\beta}(\hat{\pi}; \rho_{\text{prior}})\right]\mathbb{E}\left[D_{\text{TV}}\left(p_{\text{target}}(c_{\mathbf{k}_t}), p_{\text{prior}}(c_{\mathbf{k}_t})\right)\right] \tag{18}$$

$$+ \sqrt{2D_{\text{KL}}\left(p(c_{1:K}), p_{\mathcal{M}}\right)} + 1/M. \tag{19}$$

Since $\mathbb{E}_{\mathcal{D}_{\text{target}}}\big[R_{\pi_\beta}(\hat{\pi} \mid s_0, \ell, c_{\text{PALO}})\big] = R_{\pi_\beta}(\pi_{\text{PALO}}; \rho_{\text{target}})$, plugging Eq. (19) into Eq. (17) gives the desired result:

$$R_{\pi_\beta}(\pi_{\text{PALO}}; \rho_{\text{target}}) \leq \big[R_{\pi_\beta}(\hat{\pi}; \rho_{\text{prior}})\big]$$
$$+ \mathbb{E}\big[D_{\text{TV}}\big(p_{\text{target}}(c_{\mathbf{k}_t}), p_{\text{prior}}(c_{\mathbf{k}_t})\big)\big] + \sqrt{2D_{\text{KL}}\big(p(c_{1:K}), p_{\mathcal{M}}\big)} + 1/M$$
$$+ \frac{\sqrt{M} + \sqrt{n\log(Mn)}}{n} + 1/K + N^{-2/K}. \tag{20}$$

$\square$

*Proof of Lemma A.2.* Define $\{X_i\}_{i=1}^H$ to be the unique index $k$ such that $i \in u_k$, and $\{X_i'\}_{i=1}^H$ to be the unique index $k$ such that $i \in u_k'$. We have

$$\Pr\Big(\sum_{k=1}^K |u_k \cap u_k'| \geq H\varepsilon\Big) = \Pr\Big(\sum_{i=1}^H \mathbb{1}\{X_i = X_i'\} \geq H\varepsilon\Big)$$
$$= \Pr\Big(\sum_{i=1}^H \mathbb{1}\{X_i \neq X_i'\} \leq H(1-\varepsilon)\Big). \tag{21}$$

Now, we observe

$$\Pr(X_i \neq X_i') = \sum_{k=1}^K \big(1 - p_{X_i}(k)\big)p_{X_i'}(k) \tag{22}$$

$$= 1 - \sum_{k=1}^K p_{X_i}(k)^2. \tag{23}$$

Eq. (22) is concave in $p_{X_i}$, and so is maximized when for any $\delta p_{X_i}$ and some $\lambda$,

$$\lambda \delta p_{X_i}(k) = -2\sum_{k=1}^K p_{X_i}(k)\delta p_{X_i}(k),$$

i.e., when $p_{X_i}(k) = \text{const.} = 1/K$ for all $k$. Thus, we have

$$\mathbb{E}\big[\mathbb{1}\{X_i \neq X_i'\}\big] = \Pr(X_i \neq X_i') \leq 1 - 1/K.$$

Continuing from Eq. (21) with $\mu = \mathbb{E}\big[\sum_{i=1}^H \mathbb{1}\{X_i \neq X_i'\}\big]$,

$$\Pr\Big(\sum_{i=1}^H \mathbb{1}\{X_i \neq X_i'\} \leq H(1-\varepsilon)\Big) = 1 - \Pr\Big(\sum_{i=1}^H \mathbb{1}\{X_i \neq X_i'\} \leq \mu + (H(1-\varepsilon) - \mu)\Big)$$
$$\geq 1 - \exp\Big(\frac{-2(H(1-\varepsilon) - \mu)^2}{H}\Big) \qquad \text{(Hoeffding [70])}$$
$$\geq 1 - \exp\Big(\frac{-2H^2\big((1-\varepsilon) - (1 - 1/K)\big)^2}{H}\Big)$$
$$= 1 - \exp\big(-2H(1/K - \varepsilon)^2\big). \tag{24}$$

Taking the complement of Eq. (24) yields the desired result:

$$\Pr\Big(\sum_{k=1}^K |u_k \cap u_k'| \leq H\varepsilon\Big) \leq e^{-2H(\frac{1}{K} - \varepsilon)^2}. \tag{25}$$

$\square$

*Proof of Lemma A.3.* The statement follows from the ansatz

$$\varepsilon = \frac{1}{K} - \sqrt{\frac{\log N}{NHK}}$$

Plugging in,

$$\varepsilon + e^{-2H(\frac{1}{K} - \varepsilon)^2} = N^{-2/K} + \frac{1}{K} - \Big(\frac{\log N}{HKN}\Big)^{1/2}$$
$$\leq 1/K + N^{-2/K}.$$

$\square$

*Proof of Lemma A.4.* Recall the definition of the optimal PALO solution

$$c^* = \underset{c \in \mathcal{M}(s_0, \ell)}{\arg\min} \, R_{\text{EMP}}(c). \tag{26}$$

Now, noting regrets are bounded by 1 from Eq. (7), we have

$$\mathbb{E}_{\mathcal{D}_{\text{target}}}\big[R_{\text{EMP}}(c^*)\big] = \mathbb{E}_{\mathcal{D}_{\text{target}}}\Big[\min_{c \in \mathcal{M}(s_0,\ell)} R_{\text{EMP}}(c)\Big]$$

$$= \mathbb{E}_{\mathcal{D}_{\text{target}}}\Big[\Big(\frac{p(c)}{p_{\mathcal{M}}(c)}\Big) \min_{\{c^{(i)}\}_{i=1}^M \sim p_{c_{1:K}}} \big[R_{\text{EMP}}(c^{(i)})\big]\Big]$$

$$= \mathbb{E}_{\mathcal{D}_{\text{target}}}\Big[\min_{\{c^{(i)}\}_{i=1}^M \sim p_{c_{1:K}}} \big[R_{\text{EMP}}(c^{(i)})\big]\Big]$$

$$+ \mathbb{E}_{\mathcal{D}_{\text{target}}}\Big[\Big(\frac{p(c)}{p_{\mathcal{M}}(c)} - 1\Big) \min_{\{c^{(i)}\}_{i=1}^M \sim p_{c_{1:K}}} \big[R_{\text{EMP}}(c^{(i)})\big]\Big]$$

$$\leq \mathbb{E}_{\mathcal{D}_{\text{target}}}\Big[\min_{\{c^{(i)}\}_{i=1}^M \sim p_{c_{1:K}}} \big[R_{\text{EMP}}(c^{(i)})\big]\Big] + \mathbb{E}_{\mathcal{D}_{\text{target}}}\Big|\frac{p(c)}{p_{\mathcal{M}}(c)} - 1\Big|$$

$$\leq \mathbb{E}_{\mathcal{D}_{\text{target}}}\Big[\min_{\{c^{(i)}\}_{i=1}^M \sim p_{c_{1:K}}} \big[R_{\text{EMP}}(c^{(i)})\big] + D_{\text{TV}}\big(p(c_{1:K}), p_{\mathcal{M}}(c_{1:K})\big)\Big]$$

$$= \mathbb{E}_{\mathcal{D}_{\text{target}}}\Big[\Pr\big(R_{\text{EMP}}(c_{1:K}) < c^{(i)} \text{ for } \{c^{(i)}\}_{i=1}^M \sim p_{c_{1:K}}\big)$$

$$+ R_{\text{EMP}}(c_{1:K}) + D_{\text{TV}}\big(p(c_{1:K}), p_{\mathcal{M}}(c_{1:K})\big)\Big]$$

$$= \mathbb{E}_{\mathcal{D}_{\text{target}}}\Big[R_{\text{EMP}}(c_{1:K}) + D_{\text{TV}}\big(p(c_{1:K}), p_{\mathcal{M}}(c_{1:K})\big)\Big] + 1/M.$$

$\square$

*Proof of Lemma A.5.* We consider the empirical regret of $\tilde{\pi}$ using the true decomposition $u_{1:K}, c_{1:K} \sim p_{\text{target}}$, for $t \sim \text{Unif}(1 \dots H)$ and $\mathbf{k}_t$ defined as in Eq. (5):

$$\mathbb{E}_{\mathcal{D}_{\text{target}}}\Big[R_{\text{EMP}}(c_{1:K})\Big] = \mathbb{E}_{\mathcal{D}_{\text{target}}}\Big[\frac{1}{H\sqrt{d_A}} \sum_{\tau \in \mathcal{D}_{\text{target}}} \min_{u_{1:K}} \mathcal{J}(c_{1:K}, u_{1:K}, \tau)\Big]$$

$$= \mathbb{E}_{\mathcal{D}_{\text{target}}}\Big[\frac{1}{H\sqrt{d_A}} \sum_{\tau \in \mathcal{D}_{\text{target}}} \min_{u_{1:K}} \sum_{n=1}^K \sum_{t \in u_n} \|a_t - \tilde{\pi}(s_t, c_n)\|^2\Big]$$

$$\leq \mathbb{E}_{u_n, c_n \sim \mathcal{D}_{\text{target}}}\Big[\frac{1}{H\sqrt{d_A}} \sum_{\tau \in \mathcal{D}_{\text{target}}} \sum_{n=1}^K \sum_{t \in u_n} \|a_t - \tilde{\pi}(s_t, c_n)\|^2\Big]$$

$$\leq \frac{1}{H\sqrt{d_A}} \mathbb{E}_{p_{\text{target}}}\Big[\sum_{n=1}^K \sum_{t \in u_n} \|a_t - \tilde{\pi}(s_t, c_n)\|^2 + D_{\text{TV}}\big(p_{\text{target}}(c_n), p_{\text{prior}}(c_n)\big)\Big]$$

$$= \mathbb{E}_{u_n, c_n \sim p_{\text{prior}}}\Big[\frac{1}{H\sqrt{d_A}} \sum_{n=1}^K \sum_{t \in u_n} \|a_t - \tilde{\pi}(s_t, c_n)\|^2\Big] + \mathbb{E}\big[D_{\text{TV}}\big(p_{\text{target}}(c_{\mathbf{k}_t}), p_{\text{prior}}(c_{\mathbf{k}_t})\big)\big]$$

$$= R_{\pi_\beta}(\hat{\pi}; \rho_{\text{prior}}) + \mathbb{E}\big[D_{\text{TV}}\big(p_{\text{target}}(c_{\mathbf{k}_t}), p_{\text{prior}}(c_{\mathbf{k}_t})\big)\big].$$

$\square$

# B  Environment Details

We evaluate our method in a real-world tabletop manipulation setup. We use a 6DOF WidowX-250 robot interacting with various objects both inside and outside of our training distribution at 5 Hz. We use one 640×480 RGB camera mounted on top of the model as set up in BridgeData [5]. When computing observations we downsample images to $224 \times 224$.

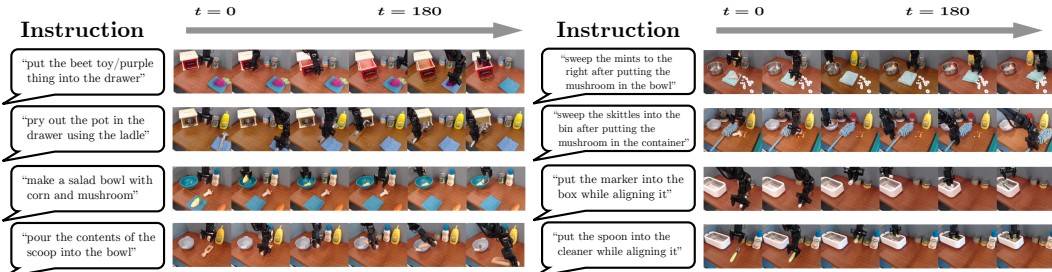

Figure 8: Sample rollouts using PALO on unseen testing tasks.

We evaluate our method in the following scenes, which include:

**Sweep:** This scene involves an object manipulation as well as sweeping task unseen in the Bridge-Data's initial training trajectories.
    **mint:** Placing the mushroom in the pot, then sweep the mints on the right using the towel.
    **skittles:** Instead of using mints and towel for sweeping, we use a swiffer and skittles instead.
**Drawer:** This scene involves using a drawer and perform manipulation within the space of the drawer.
    **put in:** Open the drawer, and then put a purple object (beet/sweet potato) inside the drawer.
    **pry away:** A pot is stored inside the drawer space, and the robot must use a ladle to pry away the pot within drawer.
**Bowl:** This scene involves object manipulation to a bowl and perform long-horizon or 6DOF manipulation.
    **salad:** This task requires sequential object manipulation by putting a corn cob and a mushroom in the bowl.
    **pouring:** This task requires the robot to grasp a scoop and pour almonds inside the scoop into the bowl.
**Rotation:** This scene involves rotating a spoon and a marker to fit into a white container not aligned with the pen/marker, and naive pick-and-place will not correctly align the object into the container.
    **spoon:** Placing the spoon in the container placed on the left side of the table.
    **marker:** Replacing the spoon with the marker and randomize location of the container while being misaligned.

We summarize the evaluation tasks in Table 1, and show example rollouts in Fig. 8.

## C   Training Details

We train on an augmented version of the BridgeDataV2 dataset [5], which features over 50k trajectories with 72k language annotations. We algorithmically augment the dataset with low-level instructions using heuristics designed over the proprioceptive states of the robot and incorporate language context by parsing the language instruction using a language model. We use the Adam optimizer [71] to minimize the loss function in Eq. (27).

Instead of naively looping through Algorithm 1, we batch our implementation with the exception of the outermost for loop, thus reducing time consumption during optimization by a significant margin via vectorization. We record an empirical time consumption of 470 seconds for our language optimization module on computations ran on a V4 TPU module, in which only 200 seconds are required for sampling 20000 different partitions to complete the optimization for all of the 15 sets of language instructions. We save our optimal plans for future use, thus reducing overhead even more.

We encode both language instructions using a frozen MUSE model [63] before passing them into the main ResNet with FiLM layers [64].

Table 1: Task Breakdown

| Scene | Task | Long-Horizon? | 6DOF required? | Instruction |
|---|---|---|---|---|
| Drawer | put in | Yes | Yes | "put the beet toy/purple thing into the drawer." |
| | pry away | Yes | Yes | "pry out the pot in the drawer using the ladle." |
| Bowl | salad | Yes | No | "make a salad bowl with corn and mushroom." |
| | pour scoop | No | Yes | "pour the contents of the scoop into the bowl." |
| Sweep | mints | Yes | No | "sweep the mints to the right after putting the mushroom in the bowl." |
| | skittles | Yes | No | "sweep the skittles into the bin after putting the mushroom in the container." |
| Rotation | marker | No | Yes | "put the marker into the box while aligning it." |
| | spoon | No | Yes | "put the spoon into the cleaner while aligning it." |

## C.1 Hyperparameter Selection

We discuss the hyperparameters used in our method and baselines.

**Policy Training** We set our learning rate for our Adam Optimizer [71] to $3 \cdot 10^{-4}$ and a dropout rate of 0.1 in our policy head. We employ random resizing and cropping, contrast, brightness, saturation, and hue for input images. We train our policy for 300,000 steps, in which we use the checkpoint with the lowest validation MSE. The total training time takes 12 hours when trained on 4 TPU pods.

**Language Decomposition Optimization** During optimization, we sample $M = 15$ random instruction sets from GPT4-o, and we use $N = 20,000$ sampling steps in order to find the best subtask decomposition.

In order to batch across demonstrations, which have different trajectory lengths, we pad our trajectories to a certain length $H$ (200 for long-horizon tasks, 150 for non long-horizon tasks). We sum the squared difference between generated action and oracle action in evaluation, thus giving a consistent error metric analogous to Eq. (7).

## C.2 Baseline Details

We finetune an Octo-small [11] model that is trained on BridgeData with a learning rate of $3 \cdot 10^{-4}$ and finetune our model's action head for 5000 steps. We use the hyperparameters set by Octo for the rest of the settings.

In order to perform tasks in long-horizon, we assign a language label for each task in order to transplant semantic understanding from human into Octo. The same language instruction for PALO evaluation is also used for Octo finetuning.

## D Augmentation Details

We train the policy by maximizing the likelihood of actions given high- and low-level instructions in the dataset $\mathcal{D}_{\text{prior}}$:

$$\mathcal{J}(\theta) = \mathbb{E}_{\mathcal{D}_{\text{prior}}} \left[ \|a_t - \pi_\theta(s_t, (c^H, \mathbf{0}))\|^2 + \|a_t - \pi_\theta(s_t, (\mathbf{0}, c^L))\|^2 + \|a_t - \pi_\theta(s_t, (c^H, c^L))\|^2 \right]$$
(27)

where $s_1 \ldots s_H \in \mathcal{S}$, $a_1 \ldots a_H \in \mathcal{A}$, $c^L, c^H \in \mathcal{L} \cup \{\mathbf{0}\}$, and $\theta$ are the parameters of the policy network, $\mathbf{0}$ is an additional point representing the absence of a high- or low-level instruction, which will be represented as an embedding vector of zero during training, and $\tau = (s_0, a_0, \ldots, s_H, a_H)$ is a trajectory sampled from the dataset.

This objective encourages the policy to learn to follow instructions at both levels of abstraction, marginalizing over missing instructions. We chunk actions within training data into segments of length 4 and evaluate the low level instruction within these segments and append them into the training data.

### D.1 Heuristics for Low-Level Language Augmentation

We algorithmically enhance our training data by using heuristics generated by the proprioception of the robot and language context, which generates the low-level instructions. The labeled language instruction is passed into a language model to obtain manipulation keywords, and we combine the keywords with the proprioceptive information within that time span including translation, rotation, and gripper movement into coherent language commands.

**Proprioception.** We use standard deviation of each action against the metadata of BridgeData [5] and determine how to describe the proprioception of the label. We determine the largest direction in which the gripper is moving (up, down, left, right, forward, backward) and the orientation it is rotating (up, down, left, right, clockwise, counterclockwise), and determine whether the movement is unambiguous enough by checking the largest z-score in translation and rotation. We then combine the movement as well as the keywords extracted to form language primitive commands.

**Target Object.** We identify the target object using a prompt heuristic to be fed into GPT3.5-Turbo [9] by taking advantage of the fact that BridgeData consists of mainly object manipulation data. We extract two keywords: the object to be manipulated and the destination of the object, based on the fact that much of BridgeData is focused on object manipulation. The precise prompt can be found at Appendix F

**Data Filtering.** We filter low-level instruction on two occasions: when the movement itself is ambiguous and when the language model gives inconsistent results. We check the former by looking up the norm of the translation and the norm of rotation, and we check the latter by using regular expression to see if the result was against the desired format and manually filtering out some common keywords of inadmissible GPT query. On the former occasion, we use an empty string as the low level instruction, and on the second occasion, we use only proprioceptive information for low-level instruction.

### D.2 Additional High-Level Language Augmentation

We additionally augment the high-level language annotations by generating context-free rephrasings with GPT-3.5 [9]. For each trajectory with crowdsourced language annotations in the BridgeData v2 dataset, we generate 5 such augmented language strings following the approach of Myers et al. [36].

## E Ablation Details

We ablate our experiment in progressive manners, going from full implementation to using only the barebone hierarchical policy network.

- PALO w/o high level instruction: while running PALO, we derive both high and low level instruction sets. However, during inference on robot, we mask out the high level instruction and feed in zero embeddings.
- PALO w/o low level instruction: mask out the low level instruction and replace them with zero embeddings during inference.

- Fixed Time During Optimization: for each trajectory that has corresponding length $H_1, H_2, \ldots, H_i$, we choose fixed $u_i = [\frac{H_i}{k}, \frac{2H_i}{k}, \ldots, \frac{(k-1)H_i}{k}]$ during optimization. We implement no $u$ sampling, which reduce PALO into an $\arg\max$ operation.
- Zero-Shot Plan Generation: instead of sampling 15 plans, we sample only one plan from VLM and examine the behavior of the robot using that specific plan.
- No VLM Guidance: We use only $\ell$ as our high level instruction, and mask out low level instruction with zero embeddings during inference.

## F  Prompting Methods

We employ a keyword decomposition prompt in our augmentation method and a planning prompt to generate VLM outputs. They are listed below:

**Keyword Decomposition Prompt**

```
User: "You are presented with a text for high level instruction for a
    robot, and you need to extract keywords in the task description
    text.
In this instruction, the first keyword is the object being moved, and
    the second keyword, if applicable, what is the moving taking this
    to (either another object or a location) within the instruction.
Only return the first and second keyword, and they should be separated
     by a comma. If the instruction is in another language, write your
     response in English.
For example, if the text instruction says "Pick up the silver lid on
    the left to the middle of two burners", return "silver lid, middle
     of two burners".
Or if the instruction says: "Move the object to the top middle side of
     the table.", your response should be "object, top middle side of
    the table".
Or if the instruction says : "Move the red greenish thing on the towel
     to the right.", return "red greendish thing on the towel, the
    right".
Try your best to find the two key phrases, but if you can't find the
    second keyword within the instruction sentence, write "N/A".
For example, if the instruction is "Move the pot lid.", the response
    should be "pot lid, N/A".
There might be some other description regarding confidence at the end,
     you are safe to ignore it.\n The specific task description for
    you to analyze is: \n {instruction} \n "
```

**Planning Prompt**

```
User: Here is an image observed by the robot in a tabletop robot
    manipulation environment. The gripper situated at the top of the
    center of table and perpendicular to it.
    Now plan for the list of subtasks and skills the robot needs to
        perform in order to {instrs}.

    Each step in the plan can be selected from the available skills
        below:

    *movement direction:
        *forward. This skill moves the robot gripper away from the
            camera by a small distance.
        *backward. This skill moves the robot gripper towards the
            camera by a small distance.
        *left. This skill moves the robot gripper to the left of the
            image by a small distance.
        *right. This skill moves the robot gripper to the right of the
             image by a small distance.
        *up. This skill moves the robot gripper upward until a safe
            height.
```

*down. This skill moves the robot gripper downward to the
    table surface.

*rotation direction:
    *left. This skill tilts the gripper to an angle to the left.
    *right. This skill tilts the gripper to an angle to the right.
    *down. This skill tilts the gripper to an angle facing up.
    *up. This skill tilts the gripper to an angle facing down.
    *clockwise. This skill rotates the gripper and the objcet it
        is holding clockwise.
    *counterclockwise. This skill rotates the gripper and the
        object it is holding counterclockwise.

*gripper movement:
    *close the gripper. This skill controls the robot gripper to
        close to grasp an object.
    *open the gripper. This skill controls the robot gripper to
        open and release the object in hand.

You may choose between using one of movement direction. rotation
    direction, or gripper movement.
If you were to choose to use movement direction, you may use one
    or two directions and include a target object, and you should
    format it like this:
"move the gripper x towards z" or "move the gripper x and y
    towards z" where x and y are the directions and z is the
    target object.
You also must start your command with "move the gripper".
    Therefore, instead of saying something like "down" or "up",
    you should phrase it like "move the gripper down" and "move
    the gripper up". Make sure to include at least one direction
    in your command since otherwise this command format won't make
     sense.

If you were to choose to use gripper movement, you should format
    the command as "close the gripper to pick up x" or "open the
    gripper to release x", where x is the target object.
You may discard the target object if necessary. In that case use "
    close the gripper" or "open the gripper".
If you think the gripper is close to the target object, then you
    must choose to use gripper movement to grasp the target object
     to maintain efficiency.

If you were to choose gripper rotation, you should format the
    command as "rotate the gripper x", where x is the target
    rotation direction. You need to make sure that in pouring
    tasks, the opening of the container is aligned with the pot.
For example, if the object is aligned vertically but you want it
    to align it horizontally, then you should call "rotate the
    gripper counterclockwise". If you want to tilt the object in
    the gripper to pour it, you should call "rotate the gripper
    left"

Pay close attention to these factors:
*Which task are you doing.
*Whether the gripper is closed.
*Whether the gripper is holding the target object.
*How far the two target objects are. If they are across the table,
     then duplicate the commands with a copy of it.
*Where the gripper is. After the end of each subtask, it is
    reasonable to assume that the gripper will not be at where it
    originally was in the image, but somewhere close to the last
    target object.

```
Especially pay attention to the actual direction between the
    gripper and the target object. Remember that the robot's angle
    is roughly the same as the camera's angle.
To determine whether the gripper should move forward or backward,
    look into the edge of the table. If the target object is
    closer to the edge of the table that is near the top of the
    image, you should move forward, and if it is closer to the
    edge that is near the bottom of the image, you should move
    backward.
At the end of each subtask, you need to use the skill "move the
    gripper back to neutral. This will move the gripper back to
    the original position of the image after completing the task.

Start by looking at what objects are in the image, and then plan
    with the direction of the objects in mind. The tasks should be
     completed sequentially, therefore you need to consider the
    position of the gripper after each task before planning the
    next task.
You should return a json dictionary with the following fields:
- subtask: this should be the key of the dictionary. It should
    contain the only the verbal description of the subtask the
    robot needs to perform sequentially in order to finish the
    task, and they should be ordered in the same way the task is
    completed.
- list of skills: this should be the value of the dictionary. It
    should be a list of skills the robot needs to perform in order
    to finish the corresponding subtask.
Be concise, and do not return any other comments other than the
    dictionary mentioned above. Do not put "subtask: " or "lsit of
     skills: " in the key and value of the dictionary you generate
    . Remember only the description and list should be returned.
```

# G    Execution Breakdown

In this section, we provide additional qualitative results for PALO.

## G.1    Inference Details

During inference, we chunk each low-level instruction into length 8 intervals, switching to the new set of low-level (and high-level, if applicable) after these 8 steps. We chose a fixed interval instead of a dynamically allocated one due to the policy choosing to mostly stay put after finishing the action prescribed by the low-level instruction.

## G.2    Success Cases

We show the full breakdowns of success cases here. Fig. 9 and Fig. 10 gives detailed description of the robot's action primitives generated by PALO during inference.

### G.2.1    Full PALO Failure

While PALO is robust in generating language primitives that help achieve the task, it does not guarantee a successful execution of the policy as shown in Fig. 11. PALO can fail when the underlying policy fails to execute a low-level motion, after which the robot may not be able to recover and complete the task.

### G.2.2    Ablation Failures

When we ablate the components of PALO, we begin to see more critical failures. Fig. 14 demonstrates a case of grounding failure when $c_H$ is masked out, i.e., when PALO loses half of the optimized task decomposition.

$\ell = $ "**pry out the pot using the ladle**"

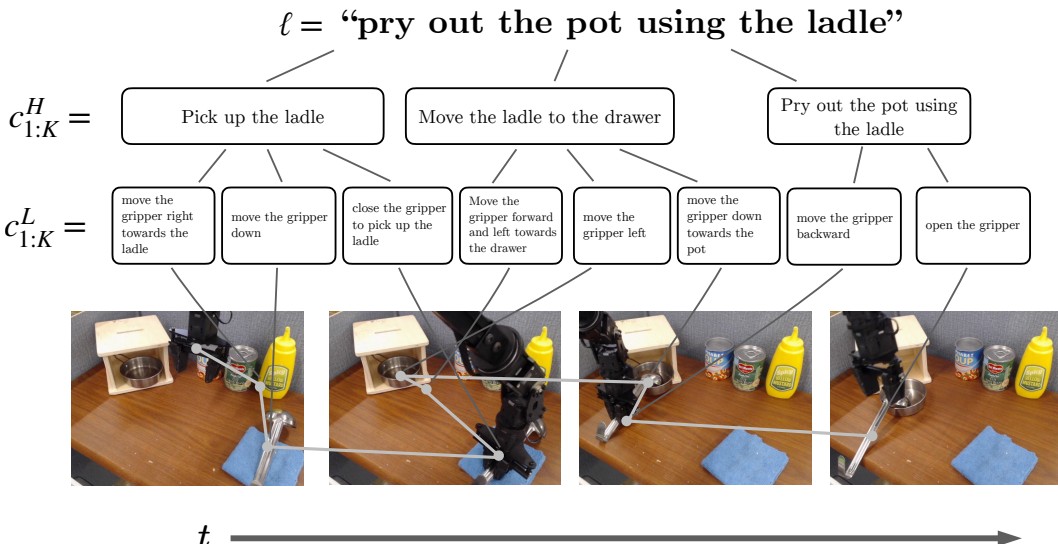

Figure 9: An execution of our method on the task "Pry out the pot using the ladle."

$\ell = $ "**pour the contents of the scoop into the bowl**"

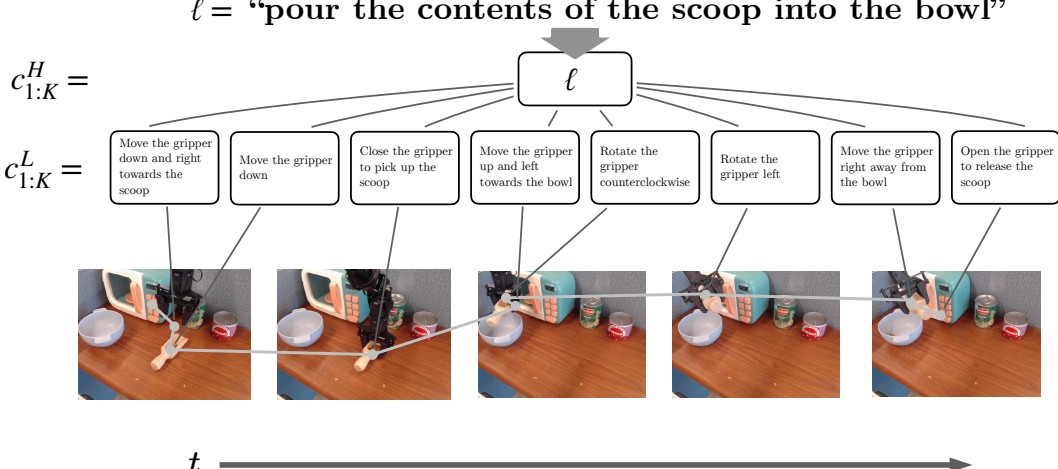

Figure 10: An execution of our method on the task "pour the contents of the scoop into the bowl.". Note that the high level instruction is $\ell$ itself, as the best-proposed language decomposition does not create additional subtasks.

Table 2: Method Comparisons

| Scene | Task | **PALO** | RT-2-X | FT-Octo | Octo | GRIF | VINN | FT-LCBC | LCBC |
|---|---|---|---|---|---|---|---|---|---|
| Drawer | put in | **0.7** | 0.0 | 0.0 | 0.2 | 0.1 | 0.0 | 0.3 | 0.1 |
| | pry away | **0.6** | 0.2 | 0.2 | 0.1 | 0.0 | 0.1 | 0.0 | 0.0 |
| Bowl | salad | **0.7** | 0.5 | 0.0 | 0.3 | 0.4 | 0.0 | 0.6 | 0.0 |
| | pour | **0.5** | 0.1 | 0.2 | 0.3 | 0.0 | 0.0 | 0.0 | 0.0 |
| Sweep | mints | **0.7** | 0.3 | 0.1 | 0.2 | 0.0 | 0.0 | 0.2 | 0.0 |
| | skittles | **0.8** | 0.4 | 0.0 | 0.4 | 0.3 | 0.0 | 0.3 | 0.2 |
| Rotation | marker | **0.9** | 0.4 | 0.0 | 0.1 | 0.3 | 0.4 | 0.4 | 0.0 |
| | spoon | **0.8** | 0.2 | 0.1 | 0.1 | 0.1 | 0.5 | 0.2 | 0.0 |
| **Average** | | **0.71** | 0.26 | 0.10 | 0.21 | 0.15 | 0.13 | 0.25 | 0.08 |

$\ell =$ **"put the turnip in the drawer"**

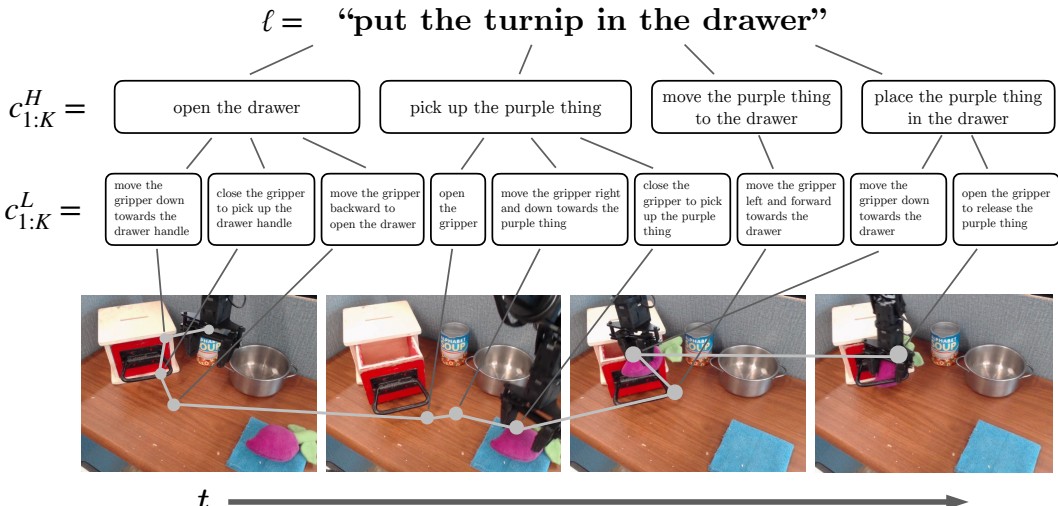

$c^H_{1:K} =$

| open the drawer | pick up the purple thing | move the purple thing to the drawer | place the purple thing in the drawer |

$c^L_{1:K} =$

| move the gripper down towards the drawer handle | close the gripper to pick up the drawer handle | move the gripper backward to open the drawer | open the gripper | move the gripper right and down towards the purple thing | close the gripper to pick up the purple thing | move the gripper left and forward towards the drawer | move the gripper down towards the drawer | open the gripper to release the purple thing |

$t \longrightarrow$

Figure 11: Failure in execution: while the robot completed every subtask correctly up until the last subtask, it did not achieve it due to errors within the policy.

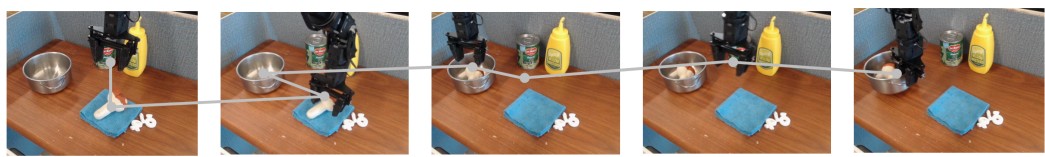

Figure 12: Spatial reasoning failure occurred when masking out low level instruction. The task was to "sweep the mints using the towel." Due to the presence of the pot and the mushroom, being both strong priors within BridgeData, the policy chose not to follow the high level instruction.

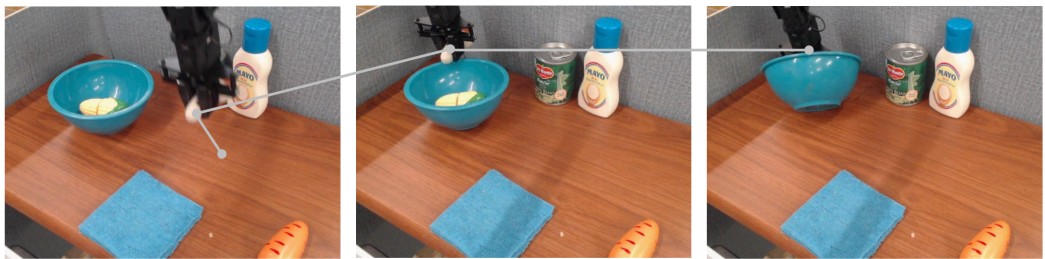

Figure 13: Grounding failure occurs when high level instruction is masked out. While the low level instruction "move the gripper left" correctly predicts the next reasonable action, masking out the context of the subtask "put the mushroom in the bowl" causes the policy to overshoot its trajectory.

Table 3: Ablations

| Scene | Task | **PALO** | No $c^H$ | No $c^L$ | Fixed Times | Zero-shot | No VLM |
|---|---|---|---|---|---|---|---|
| Drawer | put in | **0.7** | 0.2 | 0.4 | 0.4 | 0.3 | 0.0 |
| | pry open | **0.6** | 0.4 | 0.2 | 0.1 | 0.4 | 0.1 |
| Bowl | salad | **0.7** | 0.4 | 0.5 | 0.4 | 0.2 | 0.0 |
| | pour scoop | **0.5** | 0.1 | 0.4 | 0.4 | 0.2 | 0.0 |
| Sweep | mints | **0.7** | 0.5 | 0.3 | 0.5 | 0.0 | 0.0 |
| | skittles | **0.8** | 0.7 | 0.2 | 0.5 | 0.4 | 0.2 |
| Rotation | marker | **0.9** | 0.6 | 0.3 | 0.3 | 0.1 | 0.3 |
| | spoon | **0.8** | 0.6 | 0.1 | 0.2 | 0.3 | 0.2 |

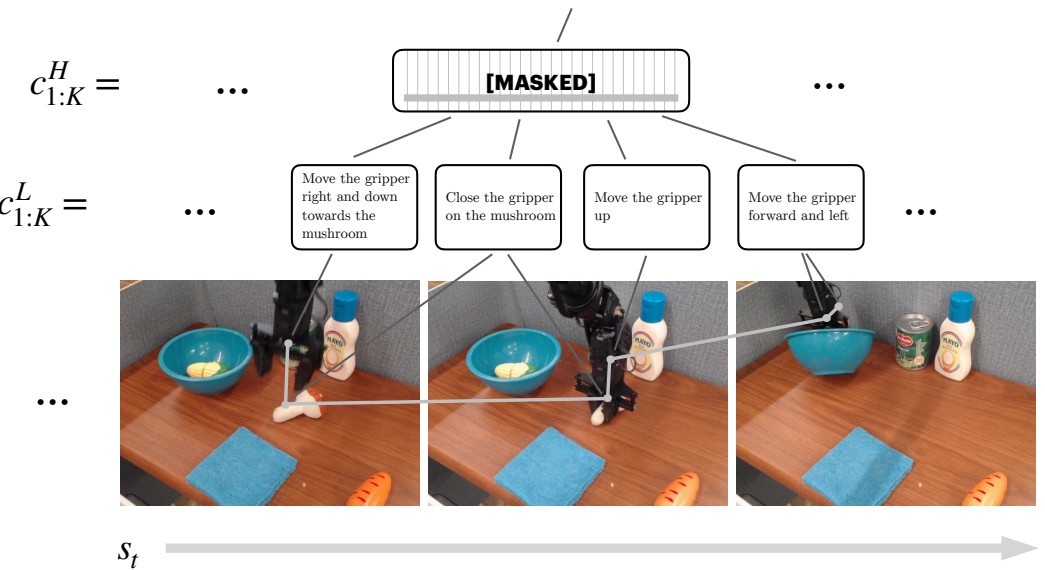

Figure 14: In this instance, we mask out the high level instructions, and the policy is only conditioned on the low-level instructions. We see that the low-level instruction "move the gripper forward and left." causes the robot to overshoot its trajectory and causes failure in execution.

## H   Evaluation Results

We present detailed results of our method across four tasks in the studied scenes in Table 2. We also present ablation results in Table 3. We evaluate each entry of the result for 10 trials, shifting the starting location of both target and background objects randomly.

## I   Code

We make our code publicly available at `https://github.com/vivekmyers/palo-robot`.

