# OpenReview forum: "Policy Adaptation via Language Optimization: Decomposing Tasks for Few-Shot Imitation"
_robot-learning.org/CoRL/2024/Conference — CoRL 2024_

### Official Review · Reviewer_AzAi · 2024-07-21
**PALO provides an intuitive method for language decomposition to aid few-shot imitation of language conditioned policies that has been explained well, thoroughly evaluated through several experiments and shows interesting results.**

**Originality:** 3
**Technical Quality:** 5
**Clarity Of Presentation:** 5
**Potential Impact:** 2
**Recommendation:** 3
**Confidence:** 4

**Review:**

Strengths

- PALO is a well written paper that is easy to follow and understand. The provided appendix provides sufficient additional details to understand the method’s implementation details.
- It addresses an important problem: that is how can we find a good decomposition of language descriptions for a task such that a pretrained policy can succeed on a previously unseen setting which will be of interest to the robot learning community
- The method is simple and intuitive making it appealing. In particular the fact that with appropriate prompting GPT4-o can provide low level language commands that focus so well on instructing the policy about what low level actions to take appears to be very powerful; especially, given that doing so appears to outperform simply fine-tuning on new demonstrations.
- The experiments are well executed and thorough. A number of good baselines were selected to compare PALO against and the performed ablations  confirm the authors’ design choices for PALO.

Weaknesses

- See questions for rebuttal

**Quality Of The Limitations Section:**

3

**Questions For Rebuttal:**

- Can you elaborate on how you created the training dataset? How did you select the right language decompositions of the tasks present such that the low level language commands are correct?
- Could you provide an experiment that studies the performance of PALO vs fine-tuning under different numbers of demonstrations ? Fine tuning OCTO for example on a very large number of demonstrations could result in better performance than PALO, it would be interesting to see when that happens in terms of task specific demonstrations.
- It would be interesting to see how would the pretrained policy perform if a random language decomposition is selected and used to fine-tune the network used for PALO on the few provided demonstrations, instead of fine-tuning Octo.
- If a failure occurs during task execution, would PALO be able to recover given that the low level instructions are fed to the policy sequentially without checking whether the low level action was successfully performed?

**Robotics Focus:**

4

**Summary Of Paper:**

The paper proposes PALO, a method that performs few-shot imitation by leveraging a language conditioned policy pre-trained on BridgeData and task-specific demonstrations to find a sequence of high + low level commands that make the pre-trained policy succeed in a task for which a human provides a few demonstrations. To achieve this, given the task specific demonstrations and language instruction, PALO uses an LLM to decompose the language instruction into lower level commands. Then, by optimising over the different possible low level commands and the timing to switch between them, PALO selects the sequence of high+low level commands that minimises the validation loss of the pre-trained policy on the provided demonstrations. This sequence of language commands is then selected and the task is performed. The authors find that prompting the policy with a good sequence of low level commands outperforms other alternatives, such as fine-tuning on the demos and through several experiments and ablations the paper shows that PALO outperforms several baselines.

**Summary Of Recommendation:**

My recommendation relies on the fact that overall the paper presents a novel method on how to decompose a task into lower level instructions to aid few shot policy learning that will be of interest to the community. The method explanation is simple and straightforward and experimental evaluations along with their ablations are convincing.

---

### Official Review · Reviewer_MWui · 2024-07-21
**Interesting method, needs more analysis and experiments to validate the claims**

**Originality:** 3
**Technical Quality:** 3
**Clarity Of Presentation:** 3
**Potential Impact:** 3
**Recommendation:** 2
**Confidence:** 3

**Review:**

### Strengths:
1. Interesting method for learning long-horizon tasks from few demonstrations by decomposing the task into subtasks using GPT4o and then learning a high and low-level conditioned policy.
2. Pretraining objective is interesting, as is conditions on both low and high level instructions, while also adapting to missing instructions in pretraining data.
3. Several baselines are included in the comparison.
4. Intuitive ablations are also performed, giving insight into the method.
5. Qualitative experiments are also shown along with failure cases, and limitation are discussed.
6. The supplementary video is good and improves understanding of the method.

### Weaknesses:
1. All experiments are missing std deviations, to ensure that the runs are not cherry-picked, several runs should be done for every expt.
2. Since this method is more of an algorithm rather than an architecture, it would much more general to show that it works with more architectures as well, like Octo, RT2X, R3M, MVP, etc.

**Quality Of The Limitations Section:**

3

**Questions For Rebuttal:**

1. How exactly is ‘u’ sequence initialized? Is it annotated by the human (sec 3.2, 3.3)?
2. Is it useful to condition the policy on s0? Why or why not?
3. How do you get c_L for the D_prior?
4. Please also include how D_prior is different from D_target, via both qualitative examples and visualizing in encoding space with methods like t-SNE. It is unclear how D_target is OoD. Also, in what sense is it unseen - is it an unseen object, scene, or skill? A plot visualizing and summarizing this information should be added.
5. Pretraining data stats are missing.

**Robotics Focus:**

4

**Summary Of Paper:**

Learning long-horizon tasks from few demonstrations by decomposing the task into subtasks using GPT4o and then learning a high and low-level conditioned policy

**Summary Of Recommendation:**

I think if the authors add required details pointed above and add the missing experiments, then the paper will be more complete.

---

### Official Review · Reviewer_15gE · 2024-07-24
**Review for Submission698**

**Originality:** 2
**Technical Quality:** 2
**Clarity Of Presentation:** 2
**Potential Impact:** 2
**Recommendation:** 3
**Confidence:** 4

**Review:**

#

### Clarity

The paper is well motivated, but some crucial details are missing or unclear:

- The exact procedure for optimizing over task decompositions is not fully explained.
- The distinction between high-level and low-level instructions in the method could be elaborated on.
- Section 4.4: The ablations performed and the information gained from each ablation is hard to follow. A clearer presentation of the ablation studies and their implications would strengthen the paper.

### Originality and Significance

The concept of using a sampling-based method for few-shot adaptation in robotics is interesting and potentially significant. While sampling methods have been shown to perform well in limited data settings in previous work, applying this approach to language-guided robotics tasks is novel.

### Strengths

1. Presents a simple, general recipe for few-shot adaptation in robotic tasks.
2. Demonstrates the potential of leveraging large language models for robot task planning.

### Weaknesses

1. Methodology: The details of the sampling method used are not clearly explained. More information on how the optimal task decomposition is selected would be beneficial.
2. Experimental details: Crucial information such as the number of demonstrations used in experiments and number of evaluations done per task is missing. This makes it difficult to fully assess the method's few-shot capabilities.
3. Baselines: The comparison is primarily against parametric methods. Including a comparison to other non-parametric methods for sampling would provide a more comprehensive evaluation, especially given that non-parametric methods have been shown to perform well in limited data settings.[1]
4. Analysis of failures: The paper would benefit from a more in-depth analysis of failure cases. For tasks such as "pry away" where the method struggles, it would be valuable to understand if these failures are due to limitations in the language embeddings generated, the image encoder used, or other factors.
5. Non parametric methods, while performant in regimes with limited data, may not scale to a large number of tasks. It would be useful to add this to the limitations section.

[1] Pari, J., Muhammad, N., Pandian, S. and Pinto, L., The Surprising Effectiveness of Representation Learning for Visual Imitation.

**Quality Of The Limitations Section:**

1

**Questions For Rebuttal:**

1. How many demonstrations were typically used in your few-shot experiments?
2. How many evaluations were run per task?
3. Can you provide more details on the optimization procedure used to select the best task decomposition? (My current understanding is that it is based on randomly sampling combinations of subtasks and finding action sequences with the lowest MSE)
4. Have you considered comparing PALO to other non-parametric methods for few-shot learning in robotics? Including comparisons with non-parametric baselines to could better contextualize the method's performance in few-shot settings.
5. Do you have any insights into what causes failures in tasks such as "pry away"? Could these be attributed to limitations in the language embeddings generated or the image encoder used?

**Robotics Focus:**

4

**Summary Of Paper:**

This paper presents PALO, a method for few-shot adaptation of language-conditioned robot policies to unseen tasks. The key idea is to leverage a VLM to generate candidate task decompositions from a sequence of frames, and then optimize over these decompositions using a small set of demonstrations. This allows the method to adapt a pre-trained policy to new tasks without fine-tuning, by selecting an appropriate sequence of subtasks that the policy can execute.

**Summary Of Recommendation:**

This paper presents an interesting approach to few-shot adaptation in language conditioned robot manipulation. Making appropriate revisions to address concerns around methodological details and experimental details, could make strengthen this paper.

---

### Author Rebuttal · Authors · 2024-08-10

We would like to thank the reviewers for their thoughtful comments. See the summary and individual comments below for detailed responses to all of the concerns raised. We have attached the revised paper here with key changes and additional results highlighted in red.

---

### Decision · Program_Chairs · 2024-09-04

**Decision:**

Accept

**Comment:**

Strengths:

- Interesting approach that uses GPT4o for decomposing tasks into subtasks and uses it to do few-shot adaptation
- Well-written paper with lots of experiments, ablations and comparisons that highlight the strengths of the work

Weaknesses:
- Some details about the demonstration dataset and pre-training are missing as pointed out by the reviewers.
- More insights and analysis of failures seen during experiments would make the paper stronger. Moreover, please provide multiple runs of experiments, along with some statistical analysis of the performance.

Post-rebuttal discussion:
In the rebuttal phase, authors provided additional details missing in the original submission, as well as baselines and ablations, including multiple real-world runs of the baselines.